# Quantum Annealing in the NISQ Era: Railway Conflict Management

**DOI:** 10.3390/e25020191

**Published:** 2023-01-18

**Authors:** Krzysztof Domino, Mátyás Koniorczyk, Krzysztof Krawiec, Konrad Jałowiecki, Sebastian Deffner, Bartłomiej Gardas

**Affiliations:** 1Institute of Theoretical and Applied Informatics, Polish Academy of Sciences, Bałtycka 5, 44-100 Gliwice, Poland; 2Wigner Research Centre, Konkoly-Thege M. út 29-33, H-1525 Budapest, Hungary; 3Faculty of Transport and Aviation Engineering, Silesian University of Technology, 40-019 Katowice, Poland; 4Institute of Physics, University of Silesia, 41-500 Chorzów, Poland; 5Department of Physics, University of Maryland, Baltimore County, Baltimore, MD 21250, USA; 6Instituto de Física ‘Gleb Wataghin’, Universidade Estadual de Campinas, Campinas 13083-859, SP, Brazil

**Keywords:** railway dispatching problem, quadratic unconstrained binary optimization (QUBO), quantum annealing

## Abstract

We are in the noisy intermediate-scale quantum (NISQ) devices’ era, in which quantum hardware has become available for application in real-world problems. However, demonstrations of the usefulness of such NISQ devices are still rare. In this work, we consider a practical railway dispatching problem: delay and conflict management on single-track railway lines. We examine the train dispatching consequences of the arrival of an already delayed train to a given network segment. This problem is computationally hard and needs to be solved almost in real time. We introduce a quadratic unconstrained binary optimization (QUBO) model of this problem, which is compatible with the emerging quantum annealing technology. The model’s instances can be executed on present-day quantum annealers. As a proof-of-concept, we solve selected real-life problems from the Polish railway network using D-Wave quantum annealers. As a reference, we also provide solutions calculated with classical methods, including the conventional solution of a linear integer version of the model as well as the solution of the QUBO model using a tensor network-based algorithm. Our preliminary results illustrate the degree of difficulty of real-life railway instances for the current quantum annealing technology. Moreover, our analysis shows that the new generation of quantum annealers (the advantage system) does not perform well on those instances, either.

## 1. Introduction

Concentrated efforts all around the globe [1,2,3,4,5] are pursuing the development of viable quantum technologies. However, the technological challenges are immense, and it may still take some time before the first fault-tolerant quantum computers may become available for practical applications [6]. Thus, it is of instrumental importance to not only build a quantum literate workforce [7] but also ensure investments are made in realistic and societally beneficial avenues for development [8].

Despite the fact that the first demonstrations of quantum advantage have been published [9], currently available hardware is still prone to noise. Thus, it has been argued that we are in the era of noisy-intermediate scale quantum (NISQ) technologies [10]. For instance, the D-Wave quantum annealer promises to deliver scalability beyond current classical hardware limitations. However, exploiting NISQ technologies often requires a different mathematical modeling framework. The D-Wave quantum annealer accepts an Ising spin-glass instance, possibly in the form of a quadratic unconstrained binary optimization (QUBO) problem equivalent to it, as its input and outputs solutions encoded in spin configurations. High-quality solutions are expected to be computed by these devices in a reasonable time, even for problems of the size which already bear practical relevance (currently, up to 5000 variables on a sparse graph [11]). More importantly, an NISQ computer may not (yet) be able to outperform classical computers; however, seeking and demonstrating amendable applications provides the instrumental guiding principle for the development of purpose-specific devices with genuine quantum advantage.

To date, at least in public domain research, most of the studied “quantum” problems are not directly relevant to a particular industrial application but rather concern the solution of “classical” generic, computationally hard problems, such as, e.g., the traveling salesman problem or the graph coloring problem [12]; see, e.g., Ref. [13] for a comprehensive review. The present work belongs to a more practically motivated line of research: it is dedicated to making quantum computing more broadly accessible by demonstrating its applicability to a relevant problem from a field not directly related to physics: conflict management in railway operation. Railway operations involve a broad range of scheduling activities, ranging from provisional timetable planning over rolling stock circulation planning, crew scheduling and rostering, etc. to operational train dispatching in case of disturbances, such as, e.g., severe weather, unplanned events, and technological breakdown. Many of these tasks require solving computationally expensive and overall challenging combinatorial problems. Various consequences of improper planing, e.g., incorrect dispatching decisions can be severe in terms of resources (e.g., time costs, passengers’ satisfaction, financial loss).

In the domain of transportation research, the applicability of quantum annealing has only been demonstrated for very few problems. For instance, Stollenwerk et al. [14] recently addressed a class of simplified air traffic management problems of strategic conflict resolution. Their preliminary results show that some challenging problems can be solved efficiently with the D-Wave 2000Q machine, see Figure 1. As for the railway industry, to the best of our knowledge, a preliminary version of the present work [15] was the first to apply a quantum computing approach to a problem in railway optimization. As the citations to our e-print illustrate [16,17], this research direction is attracting increasing interest.

The main purpose of the present paper is to elucidate how railway management problems can be solved with the currently available hardware. Naturally, we do not expect the current generation of the D-Wave annealer to outperform the best available classical algorithms. Rather, the present work is of pedagogical and instructive value as it provides an entry point for transportation research into quantum computing and demonstrates the applicability of an NISQ computer. To this end, we solve the delay and conflict management on an existing Polish railway whose real-time solution is of paramount importance for the local community.

Realizing that especially in the NISQ era, there is still a significant language barrier between foundational quantum physics and real-life applications, the present paper strives to be as introductory and self-contained as possible. In particular, Section 2 provides a brief review of railway conflict management as well as quantum annealing. Our model of the “real” problem is then outlined in Section 3 before we discuss our findings in Section 4. The discussion is concluded with a few remarks on future research directions in Section 5. In the Appendix A, we give a fully detailed description of our model. We include there also the description of a possible linear integer programming formulation that we use for comparison. The Appendix A contains a number of additional particular instances and their solutions.

## 2. Railway Dispatching Problem on Single-Track Lines

Railway dispatching problem management is a complex area of transportation research. Here, we focus on the delay management on single-track railways. This problem concerns the operative modifications of train paths upon disturbances in railway traffic. Incorrect decisions may cause the dispatching situation to deteriorate further by propagating the delay, resulting in unforeseeable consequences. Henceforth, we discuss this problem’s details and survey the relevant part of the literature. Although we focus on single-track railway lines, some considerations may also be applicable to multi-track railways [18].

### 2.1. Problem Description

Consider a part of a railway network in which the traffic is affected by a disturbance. As a result, some trains cannot run according to the original timetable. Hence, a new, feasible timetable should be created promptly, minimizing unwanted consequences of the delay. To be more specific, we are given a part of a railway network (referred to as the *network*), such as, e.g., those depicted in Figure 2a,b. The *network* is divided into *block sections* or *blocks* (This term originated in the railway signaling terminology. In general, it refers to a section of the railway line between two signal boxes.): sections which can be occupied by at most one train at a time. The block sections are labeled with numbers in the figures. We focus on single-track railway lines. These include *passing sidings* (referred to as *sidings*): parallel tracks, typically at stations; the *blocks* are labeled with upper indices in parentheses in the figures. Via the sidings, trains heading in opposite directions can meet and pass, while trains heading in the same direction can meet and overtake.

All trains run according to a *timetable*. Examples of timetables are illustrated in Figure 2c,d in the form of train diagrams, and they will be explained later in Section 4.1. The set of given time–location points of a given train are termed as the *train path*, which represents a train in a train diagram as points connected with straight lines. We assume that the initial timetable is *conflict free* and that it meets all feasibility criteria. The criteria may vary [19,20] depending on the railway network in question. The possible variants include technical requirements such as speed limits, dwell times, and other signaling-imposed requirements, as well as rolling stock circulation criteria and passenger demands for trains to meet. The railway delay management problem can be viewed from various perspectives, including that of a passenger, the infrastructure manager, or a transport operation company [19,20,21]. Here, we look at this problem from the perspective of the infrastructure manager, who is to make the ultimate decision about the modifications and is in the position to prioritize the requirements.

In what follows, we assume that—for whatever reason—a delay occurs. Hence, some trains’ locations differ from the scheduled ones. A *conflict* is an inadmissible situation in which at least two trains are supposed to occupy the same block section. For instance, if an already delayed train would continue its trip according to the original plan shifted in time with the delay while the other trains would run according to the original timetable, multiple trains could meet in the same block, as illustrated in Figure 3a.

The objective of conflict management is thus to redesign the timetable to avoid conflicts (such as in Figure 3b in our example), and minimize delays. The overall delay of a train is the sum of two types of delays. A *primary delay* is caused by an initial disturbance directly, e.g., a particular train is delayed because of an engine breakdown. Such a delay cannot be avoided. Moreover, it has additional consequences as it propagates through the *network*. To separate the primary (unavoidable) part of the overall delay from the rest, which depends on dispatching decisions, the following consideration is commonly made. Obviously, given an actual location of trains, there is a minimal amount of time needed for each train to reach further destinations, e.g., due to speed limits, even when the train would not interact with any other train. The so-calculated delay is considered as the *primary delay*.

The delay of a train beyond the *primary delay* is termed as the secondary delay. These are induced by conflicts, i.e., interactions of trains, that have to be resolved by appropriate dispatching decisions. The objective of the optimization of these decisions is the minimization of a suitable function of secondary delays, e.g., their maximum or a weighted sum. Note that there are many other practically relevant options for the objective function [22], e.g., the total passenger delay or the cost of operations, and some of these are also in line with our approach.

The mathematical treatment of railway delay and conflict management leads to NP-hard problems (In computational complexity theory, NP-hardness, non-deterministic polynomial-time hardness, is the defining property of a class of problems that are informally “at least as hard as the hardest problems in NP, that is the class of problems that can be solved in polynomial time on a non-deterministic Turing machine [23]”). Certain simple variants are NP-complete [24]. It is broadly accepted that these problems are equivalent to job-shop models with blocking constraints [25], given the release and due dates of the jobs and depending on the requirements of the model and additional constraints such as recirculation or no-wait. The correspondence between the metaphors is the following. Trains are the jobs and block sections are the machines. Concerning the objective functions, the (weighted) total tardiness or make-span is typically addressed, which is the (weighted) sum of secondary delays or the minimum of the largest secondary delay in the railway context. So, with the standard notation of scheduling theory [26], our problem falls into the class $\left(J_{m}|r_{i},d_{i},block|\sum_{j}w_{j}T_{j}\right)$. Here, $J_{m}$ stands for a job shop with multiple machines, $r_{i}$ stands for the given release times, and $d_{i}$ stands for the deadlines for each job, and block stands for the presence of blocking constraints (i.e., a job blocks a machine while it is processed). The objective, the third part of the triplet, is a total weighted tardiness.

### 2.2. Existing Algorithms

The following summary of railway dispatching and conflict management is focused on the works that are closely related to the problem addressed by us. A more comprehensive review of the huge literature on optimization methods applicable to railway management problems can be found in many related publications, notably in Refs. [20,22,27,28,29,30,31].

On a single-track line, the possible actions that can be taken to reschedule trains are the following: allocating new arrival and departure times, changing tracks and platforms, and reordering the trains by adjusting the meet-and-pass plans [20,22,32]. An important issue in modeling single-track lines is the handling of sidings (stations). A recent work of Lange and Werner [33] addressing the problem describes three approaches. In the *parallel machine approach*, it is assumed that each track within the siding corresponds to a separate machine in the job shop, thereby losing the possibility of flexible routing, i.e., changing track orders at a station afterward. In the *machine unit approach,* parallel tracks are treated as additional units of the same machine. Finally, in the *buffer approach*, the sidings at the same location are handled as buffers without internal structure and therefore not warranting the feasibility of track occupation planning at a station. We adopt the buffer approach in our model.

As to the nature of the decision variables, two major classes of models can be identified:*Order and precedence variables* prescribe the order in which a machine processes jobs, i.e., the order of trains passing a given block section in the railway dispatching problem on single-track lines.*Discrete time units*, in which the decision variables belong to discretized time instants; the binary variables describe whether the event happens at a given time.

These two approaches lead to different model structures, which are hard to compare. The *discrete time units* approach appears to be more suitable for a possible QUBO formulation, but it leads to many decision variables and thus worse scaling. On the other hand, the *order and precedence variables approach* can lead to a representation of the problem with alternative graphs [34,35], which is an intuitive picture. The solution of this problem representation leads to mixed-integer programs that can be solved, e.g., with iterative methods (such as branch-and-bound), but they are not ideal for a reformulation to QUBO. Time-indexed variables, on the other hand, can result in pure binary problems that are suitable for a transformation to QUBOs [36], so we follow the latter approach.

Returning to Ref. [33], the authors considered the problem adopting the *parallel machine approach* and the *machine unit approach* with *order and precedence variables*, addressing the problem $J_{m}\left|r_{i},d_{i},block,rcrc\right|\sum_{j}T_{j}$ in the standard notation of scheduling theory. In the case of the instances addressed in this reference, with 15 or more stations and 11 or more trains, the computational time of the presented classical algorithms is reported to be always higher than 10 min using *CPLEX 12.6.1*, IBM Armonk, New York, USA, which can be considered as a long time in a dispatching situation. These illustrate the limitations of the state-of-the-art classical algorithms.

In the present work, we will adopt slightly different constraints and objectives, namely, $J_{m}\left|r_{i},d_{i},block\right|\sum_{j}w_{j}T_{j}$. As to decision variables, we opt for discrete time units and time-indexed variables. (For the sake of completeness, in the Appendix A, we demonstrate that the problem can also be encoded with precedence variables and handled by a linear solver.)

In Ref. [37], Zhou and Zhong considered the problem of timetabling on a single-track line. The starting times of trains and their stops are given, and a feasible schedule is to be designed to minimize the total running time of (typically passenger) trains. Although their problem, notably its objective function and the input, is different, the constraints are similar to those of our problem. The authors also deal with conflicts, dwell times, and minimum headway times for entering a segment of the railway line. They handle the problem with reference to resource-constrained project scheduling. Their decision variables are the discretized entry and leave times of the trains at the track segments, binary precedence variables describing the order of the trains passing a track segment, and time-indexed binary variables describing the occupancy of a segment by a given train at a given time. They introduce a branch-and-bound procedure with an efficiently calculable conflict-based bound in the bounding step to supplement the commonly used Lagrangian approach. They demonstrate its applicability to scheduling of up to 30 passenger trains for a 24-h periodic planning horizon on a line with 18 stations in China.

Harrod [38] proposed a discrete-time railway dispatching model, with a focus on conflict management. In this work, the train traffic flow is modeled as a directed hypergraph, with hyperarcs representing train moves with various speeds. This may be confined to an integer programming model with time-, train-, and hypergraph-related variables and a complex objective function covering multiple aspects. The model is demonstrated on an imaginary single-track line with long passing sidings at even-numbered block sections of up to 19 *blocks* in length. An intensive flow of trains at moderate speeds is examined. The model instances are solved with CPLEX in the order of 1000 s of computation time. As a practical application, a segment of a busy North American mainline is used, on which the model produced practically useful results. Bigger examples were also experimented with, leading to the conclusion that the approach is promising but that it needs more specialized technology than a standard mixed-integer programming (MIP) solver to be efficient.

Meng and Zhou [39] describe a simultaneous train rerouting and rescheduling model based on network cumulative flow variables. Their model also employs discrete-time-indexed variables. They implement a Lagrangian relaxation solution algorithm and make detailed experiments showing that their approach performs promisingly on a general n-track railway network. In the introduction of their article, they tabulate numerous timetabling and dispatching algorithms.

This brief survey of the extensive literature confirms that the problem of railway dispatching and conflict management is indeed a good candidate for demonstrating new computational technology capable of solving hard problems. Only very few models have been put into practice. The size and complexity of realistic dispatching problems make it challenging for the models to solve them with current computational technology.

### 2.3. Quantum Annealing and Related Methods

Let us now turn our attention to the main tools used in the present study: quantum annealing techniques. These have their roots in adiabatic quantum computing, which is a new computational paradigm [40] that, under additional assumptions, is equivalent [41] to the gate model of quantum computation [42] (provided that the specific interactions between quantum bits can be realized experimentally [43]). Thus, this emerging technology promises to tackle complicated (NP-hard in fact [44]) discrete optimization problems by encoding them in the ground state of a physical system: the Ising spin glass model [45]. Such a system is then allowed to reach its ground state “naturally” via an adiabatic-like process [46]. An ideal adiabatic quantum computer would in this way provide the exact optimum, whereas a *real* quantum annealer is a physical device that has noise and other inaccuracies. Hence, currently existing quantum annealers are paradigmatic examples of the NISQ era [10]. Their output is only a sample of candidates that is likely to contain the optimum. Quantum annealing can be therefore regarded as a heuristic approach, which will become increasingly accurate and efficient as the technology improves.

#### 2.3.1. Ising-Based Solvers

The Ising model, introduced originally for the microscopic explanation of magnetism, has been in the center of the research interests of physicists ever since. It deals with a set of variables $s_{i}\in \left\{+1,-1\right\}$ (originally corresponding to the direction of microscopic magnetic momenta associated with spins). The configuration of *N* such variables is thus described by a vector $s\in {\left\{+1,-1\right\}}^{N}$. The model then assigns an energy to a particular configuration:(1)$$E\left(s\right)=\sum_{(i,j)\in E}J_{i,j}s_{i}s_{j}+\sum_{i\in V}h_{i}s_{i},$$
where (V,E) is a graph whose vertices *V* represent the spins, the edges *E* define which spins interact, $J_{i,j}$ is the strength of this interaction, and $h_{i}$ is an external magnetic field at spin *i*. Although the early studies of the model dealt with configurations in which the spins were arranged in a one-dimensional chain so that the coupling *J* was non-zero for nearest neighbors only, the model has been generalized in many ways, including the most general setting of an arbitrary (V,E) graph, i.e., incorporating the possibility of non-zero couplings for any i,j pair. Such a system is referred to as a spin glass in physics. For comprehensive reviews of generalizations of the Ising model, we refer to the literature [47,48].

From an operations research point of view, the physical model is interesting, since it describes a computational resource for optimization. The idea originates from the fact that in physics, the minimum energy configuration determines many properties of a material.

In mathematical programming, it is often more convenient to deal with 0–1 variables. By introducing new decision variables x∈{0,1} so that
(2)$$x_{i}=\frac{s_{i}+1}{2},$$
and the matrix
(3)$$Q_{i,i}=2\left(h_{i}-\sum_{j=1}^{n}J_{i,j}\right),\;Q_{i,j}=4J_{i,j},$$
the Ising objective in Equation (Equation 1) can be rewritten in the form of a QUBO:(4)$$\min\;x^{T}Qx,\;s.t.\;x\in {\left\{0,1\right\}}^{N}.$$
Note that the transformation in Equations (Equation 2) and (Equation 3) is actually invertible (c.f. for instance, Ref. [49]). Hence, minimizing the Ising objective in Equation (Equation 1) is *equivalent* to solving a QUBO. In what follows, we will use the QUBO form only; it can be in fact submitted as it is to the solvers and commercial quantum devices directly. Moreover, the matrix *Q* can always be chosen to be symmetric, as $Q=(Q^{\prime }+Q^{\prime T})/2$ defines the same objective. Quantum annealers accept problems both in QUBO and Ising form and provide a non-deterministic output possibly containing the solution, as it will be discussed later.

A QUBO or Ising model can be also solved with other promising techniques. With the rapid development of quantum annealing technology, probabilistic *classical* accelerators have been under massive development. In recent years, a significant progress took place in the field of programmable gate array optimization solvers (digital annealers [50]), optical Ising simulators [51], coherent Ising machines [52], stochastic cellular automata [53], and, in general, those based on memristor electronics [54].

It is therefore vital to develop modeling strategies for quadratic binary optimization and to create novel techniques for analyzing the obtained results. This should progress similarly to how the powerful solvers for linear programs first started appearing: modeling strategies for linear programs as well as sensitivity analysis had been developed ahead of the creation of the hardware.

#### 2.3.2. Quantum Annealing

An essential step in finding the minimum of an optimization problem (encoded in Equation (Equation 1)) efficiently is to map it to its quantum version. The mapping assigns a two-dimensional complex vector space to each spin, and a complete spin configuration becomes an element of the direct (tensor) products of these spaces. An orthonormal basis (ONB) is assigned to the −1 and +1 values of the variables; thus, the product of these vectors will be an ONB (called the “computational basis”) in the whole $C^{2^{N}}$. The vectors with unit Euclidean norms are referred to as “states” of the system; they encode the physical configurations. The fact that the state can be an arbitrary vector and not only an element of the computational basis means that the quantum annealer can simultaneously process multiple configurations, i.e., inherent parallelism.

As to the objective function, the spin variables are replaced by their quantum counterpart: Hermitian matrices acting on the given spin’s $C^{2}$ tensor subspace. The product of spins is meant to be the direct (tensor) product of the respective operators. Thus, the objective function Equation (Equation 1) turns into a Hermitian operator, which is referred to as the problem’s Hamiltonian:(5)$$H_{p}:=E\left({\hat{\sigma }}^{z}\right)=\sum_{\langle i,j\rangle \in E}J_{ij}{\hat{\sigma }}_{i}^{z}{\hat{\sigma }}_{j}^{z}+\sum_{i\in V}h_{i}{\hat{\sigma }}_{i}^{z},$$
whose lowest-energy eigenstate is commonly called the “ground state”. Above, $\sigma_{i}^{z}$ denotes the Pauli *z*-matrix associated with the *i*th qubit. In the present case, it is an element of the computational basis, so it represents also the optimal configuration of the classical problem. Note that the energy of a physical system is related (via eigenvalues) to a Hermitian operator, which is called its Hamiltonian. Although it seems to be a significant complication to deal with $C^{2^{N}}$ instead of having $2^{N}$ binary vectors, it has important benefits, the most remarkable of which is that they model realistic physical systems.

The main idea behind quantum annealing is based on the celebrated adiabatic theorem [55]. Assume that a quantum system can be prepared in the ground state of an initial (“simple”) Hamiltonian $H_{0}$. Then, it will slowly evolve to the ground state of the final (“complex”) Hamiltonian $H_{p}$ in Equation (Equation 5) that can be harnessed to encode the solution of an optimization problem [45]. In particular, the dynamics of quantum annealers such as D-Wave 2000Q are governed by the following time-dependent Hamiltonian [46,56]:(6)$$H\left(t\right)/\left(2\pi \hbar \right)=-g\left(t\right)H_{0}-\Delta \left(t\right)H_{p^{\prime }},\;t\in \left[0,T\right].$$
Here, the original problem’s Hamiltonian in Equation (Equation 5) must be converted into a bigger one $H_{p^{\prime }}$ whose graph is compliant with what the existing hardware can realize: the “Chimera graph” in case of DWave 2000Q; see Figure 1. The original problem’s graph will be the minor of this graph. This procedure, called “minor embedding”, is standard in quantum annealing procedures (see also Appendix A for a simple graphical representation of this *Chimera embedding*).

Many relevant optimization problems are defined on dense graphs. Fortunately, even complete graphs can be embedded into a Chimera graph [57]. There is, however, substantial overhead, which effectively limits the size of the computational graph that can be treated with current quantum annealers [58,59]. This is considered as an engineering issue that will likely be overcome in the near future [11,60]. After the Chimera embedding, the Hamiltonian describing the system reads as follows:(7)$$H_{p^{\prime }}=\sum_{\langle i,j\rangle \in E}J_{ij}^{\prime }{\hat{\sigma }}_{i}^{z}{\hat{\sigma }}_{j}^{z}+\sum_{i\in 𝒱}h_{i}^{\prime }{\hat{\sigma }}_{i}^{z},\;H_{0}=\sum_{i}{\hat{\sigma }}_{i}^{x},$$
where $\sigma_{i}^{x}$ is the *x* Pauli matrix associated with the *i*th qubit. The annealing time *T* varies from microseconds (∼2 μs) to milliseconds (∼2000 μs) depending on the specific programmable schedule [46]. As shown in Figure 1, during the evolution, g(t) varies from g(0)≫0 (i.e., all spins point in the *x*-direction) to g(T)≈0, whereas Δ(t) is changed from Δ(0)≈0 to Δ(T)≫0 (i.e., $H\left(T\right)∼H_{p^{\prime }}$). The Pauli operators ${\hat{\sigma }}_{i}^{z}$, ${\hat{\sigma }}_{i}^{x}$ describe the spin’s degrees of freedom in the *z*- and *x*-direction, respectively. Note that the Hamiltonian $H_{p}$ is classical in the sense that all its terms commute (which is the result of their multiplication, being independent of the order). Thus, as mentioned previously, its eigenstates translate directly to classical variables, $q_{i}=\pm 1$, which are introduced to encode discrete optimization problems.

The annealing time, *T* in Equation (Equation 6), is an important parameter of the quantum annealing process: it must be chosen so that the system reaches its ground state while the adiabaticity is at least approximately maintained. The adiabatic theorem gives us guidance in this respect. In the spectrum of the Hamiltonian in Equation (Equation 6), there is a difference between the energy of the ground state(s) and the energy of the state(s) just above it in energy scale. This difference is known as the (spectral) “gap”, and its minimum value in the course of the evolution determines the required computation time if certain additional conditions hold. Roughly speaking, the bigger the gap, the faster the quantum system reaches its ground state (the dependence is actually quadratic; see Ref. [61] for a detailed discussion). Thus, if the run time is not optimal, there is a finite probability of reading out an excited state instead of the true ground state.

The annealing time should be provided in advance to actually use a quantum annealer. As mentioned before, the time which would ensure that the ground state is likely to be in the resulting sample depends on the spectral gap, which is unknown. Its exact determination would be as hard as finding the actual optimum. Hence, in practice, a reasonable annealing time is determined from an educated guess, and the evolution is repeated reasonably many times, resulting in a *sample* of possible solutions (over different annealing times as well as other relevant parameters). The one with the lowest energy is considered to be the desired solution, albeit there is a finite probability that it is not the ground state. A quantum annealer is thus a probabilistic and heuristic solver. Concerning the benchmarking of quantum annealers, consult also [62].

As a side note, it should be stressed that it is not always possible to maintain the adiabatic evolution. As an example, consider the second–order phase transition phenomenon [63,64,65], in which even a short-lasting lack of adiabaticity will result in the creation of topological defects preventing the system from remaining in its instantaneous ground state. This effect, on the other hand, is a clear manifestation of the quantum Kibble–Żurek mechanism Refs. [66,67,68,69,70,71,72] and can be used to detect departures from adiabaticity. Meanwhile, a system which would evolve adiabatically in the absence of interection with its enviroment will keep evolving similarly to the ideal evolution also in certain noisy circumstances [73].

#### 2.3.3. Classical Algorithms for Solving Ising Problems

An additional benefit from formulating problems in terms of Ising-type models is that the existing methods developed in statistical and solid-state physics for finding ground states of physical systems can also be used to solve a QUBO on classical hardware. Notably, variational methods based on finitely correlated states (such as matrix product states for 1D systems or projected entangled pair states suitable for 2D graphs) have had a very extensive development in the past few decades. A quantum information theoretic insight into density matrix renormalization group methods (DMRG [74])—being the most powerful numerical techniques in solid-state physics at that time—helped with proving the correctness of DMRG. These methods also led to a more general view of the problem [75], resulting in many algorithms that have potential applications in various problems, in particular solving QUBOs by finding the ground state of a quantum spin glass. We have used the algorithms presented in Ref. [76] to solve the models derived in the present manuscript.

Neither the quantum devices nor the mentioned classical algorithms do always provide the energy minimum and the corresponding ground state (as it is not trivial to reach it [77]) but possibly another eigenstate of the problem with an eigenvalue (i.e., a value of the objective function) close to the minimum. The corresponding states are referred to as “excited states”. There are problems related to the simulations of quantum systems with NISQ quantum hardware, where only the ground state is relevant [78]. Nevertheless, excited states can also encode valuable information. This is especially apparent, for instance, in the context of neural networks where sampling is more important than finding the ground state [79]. In many optimization problems, good but not optimal solutions also bear practical relevance. Another important point in interpreting the results of such a solver is the degeneracy of the solution: it can provide multiple equivalent optima at a time.

In analyzing these optima, it is helpful that for up to 50 variables, one can calculate the exact ground states and the excited states closest to them using a brute-force search on the spin configurations with GPU-based high-performance computers. In the present work, we also use such algorithms, in particular those introduced in Ref. [80] for benchmarking and evaluating our results for smaller examples. This way, we can compare the exact spectrum with the results obtained from the D-Wave quantum hardware and the variational algorithms.

## 3. Our Model

Here, we describe our model in brief. The Appendix A provides a more detailed description. First, in Section 3.1, we encode constraints representing the railway operation scenario in the form of inequalities. To avoid continuous variables which would be incompatible with a QUBO solver, we use discretized time variables. The arising integer model is suitable to be turned into a purely 0-1 model adopting the discrete time unit approach, as described in Section 3.2. Then, in Section 3.3, the constrained 0-1 model is turned into the desired QUBO model using penalties. Finally, the QUBO model is converted to the Ising model as in Equation (Equation 1). This is completed automatically by the quantum annealer’s software package (using the binary variable transformation via Equation (Equation 2) and QUBO matrix transformation in Equation (Equation 3); the QUBO and Ising models are equivalent and any of these formulations can be submitted to solvers such as D-Wave directly). While the quantum annealer is an Ising-system inside, the solutions it returns are mapped back to the 0-1 variables of the submitted QUBO model (c.f. Equation (Equation 2)). Given such a 0-1 solution vector, it defines the actual delay of each train at each station, as described in Section 3.2. Applying these delays to the given initial timetable (with *conflicts*), a solution in the form of the *conflict free* timetable is obtained: conflicts are resolved. The presented train diagrams are constructed in this way.

### 3.1. Integer Formulation of the Constaints

Let us return to our single-track *network*: *blocks* and trains. Observe that it is only the leave times of trains from station blocks that the dispatchers decide upon, as the trains cannot meet and pass or meet and overtake on a single-track line otherwise. Let us denote the station blocks by s∈S and the set of trains by j∈J. We will formulate the problem entirely in terms of the secondary delays: $d_{s}\left(j,s\right)$ stands for the secondary delay of train *j* at the station block *s*. The detailed description in the Appendix A makes it clear that these values, along with the original timetable and the technical data (i.e., the *network* topology and time required for each train to pass a block) fully determine a modified timetable. In what follows, this description will be referred to as the *delay representation*.

In order move toward a 0–1 model, we discretize the secondary delays requiring that
(8)$$d_{s}\left(j,s\right)\in N,\;0\leq d_{s}\left(j,s\right)\leq d_{\max}\left(j\right),$$
where a reasonable upper bound $d_{\max}\left(j\right)$ can be obtained from some fast heuristics, and the time is measured in integer minutes from now on, which is a suitable scale for railway problems. When formulating constraints, it is better to work with the actual (discretized) delay $d\left(j,s\right)=d_{s}\left(j,s\right)+d_{U}\left(j,s\right)$ of train *j* at station block *s*, where $d_{U}\left(j,s\right)$ stands for the primary (unavioidable) delay. At this stage, we have defined a set of potential decision variables with finite ranges that already facilitate the formulation of a linear model for the problem, as shown in the Appendix A.

As to the constraints, we consider the following ones, which cover the requirements of the particular railway operator. In the Appendix A, we describe them in more detail while here, we give a brief summary:

**The minimum passing time condition** ensures that no block sections are passed by any train faster than allowed:(9)$$d\left(j,\rho_{j}\left(s\right)\right)\geq d\left(j,s\right)-\alpha \left(j,s,\rho_{j}\left(s\right)\right).$$
where $\rho_{j}\left(s\right)$ stands for the station block section succeeding *s* in train *j*-s sequence, while $\alpha (j,s,\rho_{j}\left(s\right))$ is the largest reserve the train can achieve by passing the *blocks* following *s* up to and together with the next station block $\rho_{j}\left(s\right)$ possibly faster than originally planned. The α values can be calculated in advance.

**The single-block occupation** ensures that at most one train can be present in a block section at a time.
(10)$$\begin{array}{cc} d(j^{\prime },s) & \geq d\left(j,s\right)+\Delta \left(j,s,j^{\prime },s\right)\\ \; & +\tau_{\left(1\right)}\left(j,s,\rho_{j}\left(s\right)\right). \end{array}$$
where Δ is the difference of the leave times of two trains from the given *blocks*, whereas $\tau_{\left(1\right)}$ is the minimum time for train *j* to give way to another train going in the same direction in the route $s\to \rho_{j}\left(s\right)$. This condition is to be tested in this form for the pair of trains $\left(j,j^{\prime }\right)$ if *j* leaves *s* before $j^{\prime }$, i.e., $d\left(j^{\prime },s\right)\geq d\left(j,s\right)+\Delta \left(j,s,j^{\prime },s\right)$; otherwise, it has to be applied so that the order of the trains is reversed.

**The deadlock condition** ensures that no pairs of trains heading in the opposite direction will be waiting for each other to pass the same *blocks*:(11)$$d\left(j^{\prime },\rho_{j}\left(s\right)\right)\geq d\left(j,s\right)+\Delta \left(j,s,j^{\prime },\rho_{j}\left(s\right)\right)+\tau_{\left(2\right)}\left(j,s,\rho_{j}\left(s\right)\right),$$
where $\tau_{\left(2\right)}\left(j,s,\rho_{j}\left(s\right)\right)$ is the minimum time required for train *j* to get from station block *s* to $\rho_{j}\left(s\right)$. Similarly to the previous condition, Equation (Equation 11) is to be applied for the pairs of trains $\left(j,j^{\prime }\right)$ so that $j^{\prime }$ is supposed to leave the block $\rho_{j}\left(s\right)$ after the train *j* leaves *s*; otherwise, the order of the trains is reversed.

**The rolling stock circulation condition** ensures the minimal technological time $R(j,j^{\prime })$ for a given train set arriving as train *j* at its terminating station $s_{j,\mathrm{end}}$ before operating again as train $j^{\prime }$:(12)$$d\left(j^{\prime },1\right)>d\left(j,s_{j,\mathrm{end}-1}\right)-R\left(j,j^{\prime }\right).$$
Certainly, this condition has to hold for pairs of trains $\left(j,j^{\prime }\right)$ which are operated with the same train set according to the rolling stock circulation plan.

There is an additional constraint: the capacity condition that could be also addressed; this would implement the buffer approach in our model. This is described in the Appendix A, but we omit it here, as we will not include it in the calculation. It would increase additional complexity that would make our model intractable with current quantum hardware, so we opted for the verification of the solutions against this condition, thereby implementing the buffer approach. Having described the constraints, now, we formulate the model as a 0-1 program and define the objective function.

### 3.2. 0-1 Formulation

To turn our model into a 0-1 problem, we introduce our final decision variables
(13)$$x_{s,j,d}=\left\{\begin{array}{cc} 1, & d(j,s)=d\\ 0, & \mathrm{otherwise} \end{array}\right.,$$
which take the value of 1 if the train *j* leaves station block *s* at delay *d* and zero otherwise. In this way, we have 0-1 variables with indices from a finite set. Observe that for constant $d_{\max}$, the number of variables is proportional to the number of trains and the number of stations minus one (as we do not consider departing from the last station in our model).

As for the objective function, we opt for a weighted sum of delays:(14)$$f\left(x\right)=\sum_{j\in J}\sum_{s\in S_{j}^{*}}\sum_{d\in A_{j,s}}f\left(d,j,s\right)\cdot x_{j,s,d},$$
where f(d,j,s) are the weights. Here, $S_{j}^{*}=S_{j}\setminus \left\{s_{j,\mathrm{end}}\right\}$, where $S_{j}$ stands for a sequence of station blocks the train runs through, $s_{j,\mathrm{end}}$ stands for the last station of *j*, and $A_{j,s}$ is the respective range of delays. It is easy to see that the weights f(d,j,s) can be chosen so that they depend on the secondary delays only; consult the Appendix A for details.

As for the constraints, let us first assume that each train leaves each station block once and only once (recall that we do not allow for recirculation):(15)$$\forall_{j}\forall_{s\in S_{j}}\sum_{d\in A_{j,s}}x_{s,j,d}=1.$$
The other constraints can be dealt with as follows.

**The minimum passing time condition** defined in Equation (Equation 9) becomes
(16)$$\forall_{j}\forall_{s\in S_{j}^{**}}\sum_{d\in A_{j,s}}\left(\sum_{d^{\prime }\in D\left(d\right)\cap A_{j,\rho_{j}\left(s\right)}}x_{j,s,d}x_{j,\rho_{j}\left(s\right),d^{\prime }}\right)=0,$$
where $D\left(d\right)=\{0,1,\ldots ,d-\alpha \left(j,s,\rho_{j}\left(s\right)\right)-1\}$, and $S_{j}^{**}=S_{j}\setminus \left\{s_{j,\mathrm{end}},s_{j,\mathrm{end}-1}\right\}$.

**The single-block occupation condition** from Equation (Equation 10) follows that
(17)$$\forall_{\left(j,j^{\prime }\right)\in J^{0}\left(J^{1}\right)}\forall_{s\in S_{j,j^{\prime }}^{*}}\sum_{d\in A_{j,s}}\left(\sum_{d^{\prime }\in B\left(d\right)\cap A_{j^{\prime },s}}x_{j,s,d}x_{j^{\prime },s,d^{\prime }}\right)=0,$$
where $B\left(d\right)=\{d+\Delta \left(j,s,j^{\prime },s\right),d+\Delta \left(j,s,j^{\prime },s\right)+1,\ldots ,d+\Delta \left(j,s,j^{\prime },s\right)+\tau_{\left(1\right)}\left(j,s,\rho_{j}\left(s\right)\right)-1\}$ is a set of delays that violates the block occupation condition.

**The deadlock condition** is to be addressed for two trains heading in the opposite direction; from Equation (Equation 11), it follows that
(18)$$\forall_{j\in J^{0}\left(J^{1}\right),j^{\prime }\in J^{1}\left(J^{0}\right)}\forall_{s\in S_{j,j^{\prime }}^{*}}\sum_{d\in A_{j,s}}\left(\sum_{d^{\prime }\in C\left(d\right)\cap A_{j^{\prime },\rho_{j}\left(s\right)}}x_{j,s,d}x_{j^{\prime },\rho_{j}\left(s\right),d^{\prime }}\right)=0$$
where $C\left(d\right)=\{d\left(j,s\right)+\Delta \left(j,s,j^{\prime },\rho_{j}\left(s\right)\right),d\left(j,s\right)+\Delta \left(j,s,j^{\prime },\rho_{j}\left(s\right)\right)+1,\ldots ,d\left(j,s\right)+$$\Delta \left(j,s,j^{\prime },\rho_{j}\left(s\right)\right)+\tau_{\left(2\right)}\left(j,s,\rho_{j}\left(s\right)\right)-1\}$, and $J^{0}$, $J^{1}$ are explained in Appendix A. In Equations (Equation 17) and (Equation 18), each train is compared with a limited number of trains; this limit is imposed indirectly by fixed $d_{\max}$. Hence, as a rough estimate of the number of terms in these two equations (for fixed $d_{\max}$), one can claim that it is proportional to the number of trains times the number of stations minus one.

**The rolling stock circulation condition** is defined in Equation (Equation 12) and can be rewritten as
(19)$$\forall_{j,j^{\prime }\in \mathrm{terminal}\;\mathrm{pairs}}\sum_{d\in A_{j,s_{\left(j,end-1\right)}}}\sum_{d^{\prime }\in E\left(d\right)\cap A_{j^{\prime },1}}x_{j,s_{\left(j,end-1\right)},d}\cdot x_{j^{\prime },s_{\left(j{,}^{\prime }1\right)},d^{\prime }}=0,$$
where $E\left(d\right)=\{0,1,\ldots ,d-R\left(j,j^{\prime }\right)\}$; this condition applies only for one station and a few selected trains.

The objective function in Equation (Equation 14) together with the constraints in Equations (Equation 15)–(Equation 19) comprise a quadratic constrained 0-1 formulation of our model. As an estimate of the number of variables and quadratic terms from Equations (Equation 13)–(Equation 18), one can conclude that (for fixed $d_{\max}$), these are proportional to n.o.trains·(n.o.stations−1).

### 3.3. QUBO Formulation: Penalties

Having formulated our problem as a constrained 0-1 program, we need to make it unconstrained to achieve a QUBO form—see Equation (Equation 4). This is usually completed with penalty methods [81]. It has been shown in [49] that all binary linear and quadratic programs translate to QUBO along some simple rules. (An alternative, symmetry-based approach [82] to constrained optimization has also been proposed in which the adiabatic quantum computer device is supposed to use a tailored $H_{0}$ term in its dynamics of the model in Equation (Equation 7). As such a modification of the actual device is not available to us, we remain using penalty methods.)

The problems one faces with a quadratic 0-1 program require certain specific considerations when adopting the penalty method. Let us outline this approach with a focus on our problem. As we have a linear objective function Equation (Equation 14), it can be written as a quadratic function because the decision variables are binary:(20)$$\min_{x}f\left(x\right)=\min_{x}c^{T}x=\min_{x}x^{T}diag\left(c\right)x.$$
(A general QUBO can contain linear terms as well; however, the solver implementations accept a single matrix of quadratic coefficients [83], so transforming linear terms into quadratic ones is more a technical than a fundamental step.)

The constraints set out in Equations (Equation 16)–(Equation 19) have to be met for a feasible solution: they are *hard constraints*. To obtain an unconstrained problem, we define a penalty function in the following way. We add the magnitude of the constrains’ violation, multiplied by some well-chosen coefficient, to the objective function.

In particular, from Equations (Equation 16)–(Equation 19), we shall have quadratic constraints in the form of
(21)$$\sum_{\left(i,j\right)\in V_{p}}x_{i}x_{j}=0,$$
excluding pairs of variables that are simultaneously 1. We can deal with such a constraint by adding to our objective the following terms:(22)$$P_{\mathrm{pair}}\left(x\right)=p_{\mathrm{pair}}\sum_{\left(i,j\right)\in {𝒱}_{p}}\left(x_{i}x_{j}+x_{j}x_{i}\right),$$
where $p_{\mathrm{pair}}$ is a positive constant. Additionally, from Equation (Equation 15), we have additional hard constraints in the form of:(23)$$\forall V_{s}\;\sum_{i\in V_{s}}x_{i}=1.$$
These constraints yield a linear expression that can be transformed into the following quadratic penalty function:(24)$$P_{\mathrm{sum}}^{\prime }\left(x\right)=\sum_{V_{s}}p_{\mathrm{sum}}{\left(\sum_{i\in V_{s}}x_{i}-1\right)}^{2}.$$
Next, we replace the $x_{i}$s with $x_{i}^{2}$s in the linear terms and omit the constant terms, as they provide only an offset to the solution, yielding:(25)$$P_{\mathrm{sum}}\left(x\right)=\sum_{V_{s}}p_{\mathrm{sum}}\left(\sum_{i,j\in V^{\times 2},i\neq j}x_{i}x_{j}-\sum_{i\in V_{s}}x_{i}^{2}\right).$$
So, our effective QUBO representation is
(26)$$\min_{x}f^{\prime }\left(x\right)=\min_{x}\left(f\left(x\right)+P_{\mathrm{pair}}\left(x\right)+P_{\mathrm{sum}}\left(x\right)\right),$$
which can be written in the form of Equation (Equation 4). We shall have many constraints similar in form to Equations (Equation 21) and (Equation 23), so we have one summed for each constraint in the objective. (It would also be possible to assign a separate coefficient to each of the constraints.)

Recall that in the theory of penalty methods [81] for continuous optimization, it is known that the solution of the unconstrained objective will tend to a feasible optimal solution of the original problem as the multipliers of the penalties ($p_{\mathrm{sum}}$ and $p_{\mathrm{pair}}$ in our case) tend to infinity, provided that the objective function and the penalties obey certain continuity conditions. As in our case, both the objective and the penalties are quadratic, and this convergence would be warranted for the continuous relaxation of the problem. Even though we have a 0-1 problem, if we had an infinitely precise solution of the QUBO, increasing the parameters would result in convergence to an optimal feasible solution.

However, somewhat similarly to the continuous case (in which the Hessian of the unconstrained problem diverges as the parameters grow, making the unconstrained problem numerically ill-conditioned), the properties of the actual computing approach or device make it more cumbersome to make a good choice of multipliers.

The parameters $p_{\mathrm{sum}}$ and $p_{\mathrm{pair}}$ have to be chosen so that the terms representing the constraints in this energy do not dominate the original objective function. If the penalties are too high, the objective is just a too small perturbation, which will be lost in the noise of the physical quantum computer or in the numerical errors of an algorithm modeling it. If, however, the penalty coefficients are too low, we obtain infeasible solutions. In the ideal case, there is a “feasibility gap” in the spectrum of solutions.

The multipliers can be assigned in an ad hoc manner by experimenting with the solution; however, a systematic, possibly problem-dependent approach to their appropriate assignment (as in the case of classical penalty methods; see [81]) would be highly desirable in order to make the QUBO more reliable and prevalent. The goal is to construct the QUBO representation in such a way that the energy landscape of the original problem is preserved. In particular, QUBO solutions that map to the feasible solutions of the original problem are expected to have lower energy than the infeasible ones. There are certain systematic methods for this: for instance, in case of linear constraints only, it is always possible to find the optimal penalty terms to separate between feasible and infeasible configurations [84]. In the present case study, we will determine penalties in an ad hoc manner based on numerical experience with the objective values and constraint violations.

Having a QUBO representation of the problem at hand (as well as an analogical Ising representation), let us turn our attention to the actual instances of our model and the results obtained for them.

## 4. Results

In this section, we discuss certain possible situations in train dispatching on the railway lines managed by the Polish state-owned infrastructure manager *PKP Polskie Linie Kolejowe S.A.* (*PKP PLK* in what follows). In particular, we consider two single-track railway lines:Railway line No. 216 (Nidzica–Olsztynek section);Railway line No. 191 (Goleszów–Wisła Uzdrowisko section).

Railway line No. 216 is of national importance. It is a single-track section of the passenger corridor Warsaw–Olsztyn, which has recently been modernized. There are both *Inter-City* (*IC*) and regional trains operating on the Nidzica–Olsztynek section of line No. 216. In this paper, we consider an official train schedule (as of April, 2020). The purpose of the analysis in this section is to demonstrate the application of our methodology to a real-life railway section.

Railway line No. 191 is of local importance. The main train service on the No. 191 railway line is Katowice–Wisła Głebce, which is operated by a local government-owned company called *“Koleje Ślaskie”* (in English, Silesian Railways; abbreviated *KS*). There are a few *Inter-City* trains of higher priority there as well. Since 2020, the traffic at this section has been suspended due to comprehensive renovation works (a temporary rail replacement bus service is in operation). Our problem instances are based on the planned parameters of the line after its commissioning based on public procurement documents [85]. On the basis of these parameters, a cyclic timetable has been created. The aim of analyzing this case is to show the broader application possibilities of the methodology.

### 4.1. The Studied Network Segment

In Figure 2a, we present a segment of railway line No. 216 (Nidzica–Olsztynek section), and in Figure 2c, the analyzed part of the real timetable is depicted in the form of a train diagram.

In Figure 2a, three stations are presented. Block 1 represents Nidzica station, which has two platform edges numbered according to the rules of *PKP PLK*. Block 3 represents Waplewo station, with another two platform edges. Olsztynek station, with three platform edges, is represented by block 5. The model involves two line blocks with the labels 2 and 4. It is assumed that it takes the same amount of time to pass through a given station block regardless of which track the train uses. To leave the station, it is required that the subsequent block is free.

As to the trains, Figure 2c represents their planned paths. Three trains are modeled: the two *Inter-City* trains in red and the regional train in black. The scheduled meet-and-pass situations take place in Waplewo and Olsztynek (which might change in case of a delay). IC5320 leaves station block 5 (Olsztynek) at 13:54, has a scheduled stopover at station block 3 (Waplewo) from 14:02 to 14:10 to meet and pass IC3521, and finally arrives at station block 1 (Nidzica) at 14:25. As to the opposite direction, IC3521 leaves station block 1 at 13:53, stops at station block 3 from 14:08 to 14:10, and arrives at station block 5 (Olsztynek) at 14:18. These two trains depart at the same time from station block 3 in opposite directions. The third train considered is R90602. It is scheduled to leave block 5 at 14:20 and stops at station block 3 (Waplewo) from 14:29 to 14:30, so it is scheduled to start occupying this track 19 min after both ICs left. It is behind IC5320 during the whole section and does not meet the IC train at all, so the original schedule is feasible and conflict free.

Now, let us add a 15-min delay to the departure time of IC5320 from station block 5 and 5-min delay to that of IC3521 from station block 1. The passing times were originally scheduled according to the maximum permissible speeds. The minimum waiting times at all the considered stations are 1 min regardless of the train type. This introduces the following situation: the two *Inter-City* trains and the regional train have a conflict at line block 4. This schedule will be referred to as the “conflicted diagram”—see Figure 3a. The resolution of this conflict requires making a decision at station blocks 3 and 5.

Let us now turn our attention to the other example. The line segment (a part of railway line No. 191) is presented in Figure 2b, while the considered train paths of the real timetable are shown in Figure 2d. There are four stations and another three stops for the passengers modeled. Block 1 represents Goleszów station, which has four platform edges. Block 2 represents a line block between Goleszów station and Ustroń station (which has two platform edges and is represented by block 3). Subsequently, we have three line blocks numbered 4, 5, and 6, with two stops for passengers: Ustroń Zdrój and Ustroń Poniwiec (with one platform edge each). Next, we have station block 7—Ustroń Polana, which has two platform edges. Between this station and Wisła Uzdrowisko station (numbered 10 with three platform edges), there are two more line blocks (8 and 9) with one stop for passengers (Wisła Jawornik). We assume that it takes exactly the same time to pass through a block regardless of the track used.

There are six trains, two *Inter-City* trains in red and four regional (*KS*) trains in black, as presented in Figure 2d. The regional trains serve all the stops and stations, while the *Inter-City* service stops only at stations. We consider Wisła Uzdrowisko (station block 10) to be a terminus for the *Inter-City* trains (however, it does not apply to the regional trains, which go farther). In this situation, there are no meet-and-pass situations at intermediate stations (Ustroń and Ustroń Polana) in the original timetable. Both *Inter-City* trains are served by the same train set, and the minimum service time is $R(j,j^{\prime })=20$ min at the terminus for ICs (block 10); see Condition SI.4 of the Appendix A.

We analyze the following dispatching cases, which have been selected to demonstrate the algorithm behavior in various situations:1.A moderate delay of the *Inter-City* train setting off from station block 1; see Appendix A.2.A moderate delay of all trains setting off from station block 1; see Appendix A.3.A significant delay of some trains setting off from station block 1; see Appendix A.4.A large delay of the *Inter-City* train setting off from station block 1; see Appendix A.

The conflicted timetables of cases 1–4 are presented in Appendix A.

### 4.2. Simple Heuristics

In the railway practice, conflicts are often resolved using simple heuristics: the First Come First Served (FCFS) and the First Leave First Served principles. A more complex one is AMCC (avoid maximum current $C_{\max}$) [34]. All of these heuristics are used to determine the order of trains when passing the blocks. In FCFS and FLFS, the way is given to the train that first arrives—or first leaves—the analyzed block section. In practice, the decisions based on both these heuristics are taken starting from the most urgent conflict. Next, since passing and overtaking is possible only at stations, so-called *implied selections* [35] are determined. The procedure is repeated as long as all the conflicts are solved.

The AMCC is a more complex approach whose objective is to minimize the maximum secondary delay of the trains; this objective will be referred to as the “AMCC objective” in what follows. This is an intuitive procedure yet more sophisticated than FCFS and FLFS. To facilitate the comparison, stations are assigned an infinite capacity. Of course, solutions requiring a capacity higher than that of the given station must be rejected.

In the example presented in Figure 2a, for the conflicted timetable in Figure 3a, each of the heuristics returns the same solution; this is presented in Figure 3b. When comparing the FCFS with the FLFS, observe that in the conflicted timetable, three trains (IC5320, IC3521, R90602) are scheduled to occupy the block 4 simultaneously, which is forbidden.

To avoid the conflict, IC3521 is allowed to enter this block with a 3-min delay at 14:17 (as soon as IC5320 leaves the block), thus leaving the block at 14:25 instead of 14:22, which results in 3 min of secondary delay. Consequently, R9062 is allowed to enter the block not earlier than 14:25, which is an additional 4-min delay as compared with the conflicted timetable. Thus, the maximum secondary delay is 4 min, and the sum of the delays on entering the last block is 7 min. The maximum secondary delay is 4 min; it is the lowest possible one, so the solution is optimal with respect to the AMCC objective. Note that in case of such a simple situation, the use of the heuristics is very close to the enumeration and evaluation of all the possible solutions, and picking the best of these. This small example is included in order to have an instance that can be fully followed manually.

The other example—the disruptions of this presented in Figure 2b—is more complex, yet it is still solvable by a state-of-the-art quantum annealer. We do not discuss this example in detail; we only present the maximum secondary delay values in Table 1 for the discussed heuristics. Recall that we need an upper bound on the secondary delays to formulate our model; we opt for $d_{max}=10$ on the basis of these data. The respective train diagrams are presented in Appendix A.

The values of the AMCC objective function are presented in Table 1; AMCC appears to find the optimum in these cases, thus providing a good enough reference for comparisons, albeit with an objective function different from that of ours. Our choice of the objective will be more flexible, thus leaving room for further non-trivial optimization.

### 4.3. Quantum and Calculated QUBO Solutions

Our QUBO approach uses the objective function set out in Equation (Equation 26). This contains the feasibility conditions (*hard constraints*) and the objective function f(x) of Equation (Equation 27). For the feasibility part, we need to determine $\tau_{\left(1\right)}\left(j,s\right)$, the minimum time for train *j* to give way to another train going in the same direction, and $\tau_{\left(2\right)}\left(j,s\right)$, the minimum time for train *j* to give way for the another train going in the opposite direction (see Appendix A).

As noted before, the QUBO objective function introduces flexibility in choosing the dispatching policy by setting the values of the penalty weights for the delays of the trains. In this way, almost any train prioritization is possible. To demonstrate this flexibility, we make the penalty values proportional to the secondary delays of the trains that enter the last station block. This is equivalent to the secondary delay on leaving the penultimate station block. Each train is assigned a weight $w_{j}$, yielding the form of Appendix A.
(27)$$f\left(x\right)=\sum w_{j}\cdot \frac{d\left(j,s^{*}\right)-d_{U}\left(j,s^{*}\right)}{d_{\max}\left(j\right)}\cdot x_{j,s^{*},d},$$
where the sum is taken over j∈J and $d\in A_{j,s^{*}}$ with $s^{*}=s_{\left(j,\;\mathrm{end}-1\right)}$. Note that this objective coincides with that of the linear integer programming approach; see Appendix A, which will be used for comparisons.

The following train prioritization is adopted. In the case of railway line No. 216, the *Inter-City* trains are assumed to have a higher value of the delay penalty weight w=1.5, while the regional train is weighted w=1.0. We assign the higher priority to the *Inter-City* train in accordance with the train prioritization rules in Poland (and in many other countries). In the other case (line No. 191), the priorities of trains heading toward block 10 (Wisła Uzdrowisko) are lower, weighted 0.9 for all the other trains in this direction. However, train priorities for the trains heading in the opposite direction (toward block 1—Goleszów—and beyond the analyzed section) have higher values: 1.0 for the regional trains and 1.5 for the *Inter-Cities*. Such a policy is motivated by the reluctance of letting the delays propagate across the Polish railway network—that regional trains proceed toward the main railway junction in the region’s major city (Katowice) and that the *Inter-City* train service is scheduled toward the state’s capital city (Warsaw). Observe that $w_{j}$ is the highest possible penalty for the delay of a train *j*; see Equation (Equation 27). In both these cases, the maximum of $w_{j}$ is 1.5. Hence, the penalties for a infeasible solution should be higher (as discussed in Section 3.3. We set $p_{\mathrm{pair}}=p_{\mathrm{sum}}=1.75>1.5$.

As mentioned before, the maximum secondary delay parameter $d_{\max}$ (which is assumed to be the same for all trains and all analyzed station blocks for sake of simplicity) cannot be smaller than the delay value returned by the AMCC heuristics. However, since the AMCC may not be optimal in terms of our objective function, we need to leave a margin for some larger values of the maximum secondary delay. On the other hand, since the system size grows with $d_{\max}$, it must be limited enough to make the problem applicable to state-of-the-art quantum devices and classical algorithms motivated by them. Specifically, since we do not analyze the delays at the last station of the analyzed segment of the line, the required number of qubits will be approximately $\left(\mathrm{number}\;\mathrm{of}\;\mathrm{station}\;\mathrm{blocks}-1\right)\cdot \left(\mathrm{number}\;\mathrm{of}\;\mathrm{trains}\right)\cdot \left(d_{\max}+1\right)$.

In the case of railway line No. 216, we set $d_{\max}=7$, which is considerably larger than the AMCC solution. There are 48 logical quantum bits needed to handle this problem instance, making it suitable for both quantum annealing at the current state of the art, and the GPU-based implementation of the brute-force search for the low-energy spectrum (ground state and subsequent excited state) [80], which is possible with up to 50 quantum bits. The benefit of this possibility is that it provides an exact picture of the spectrum, which can be used as a reference when evaluating the heuristic results of approximate methods (tensor networks) or quantum annealing. This may guide the understanding of the results of the bigger instances, in which the brute-force exact search is not available.

There are many possible distinct solutions in the case of line No. 191, making the analysis more interesting from the dispatching point of view. We set $d_{\max}=10$: for a justification, see Table 1, and observe that $d_{\max}$ is considerably larger than the AMCC output. The $d_{\max}=10$ yields 198 logical quantum bits, which we were able to embed into a present-day quantum annealer, the D-Wave device DW-$2000Q_{5}$, in most cases.

Recall that current quantum annealing devices are imperfect and often output excited states. The clue of our approach is that the excited states (e.g., returned by the quantum annealer) still represent the optimal dispatching solutions, provided that their corresponding energies are relatively small. The reason for this is that what really needs to be determined is the order of trains leaving from each station block (i.e., this is the decision to be made). What is crucial here is to determine all the meet-and-pass and the meet-and-overtake situations (in analogy with the determination of all the precedence variables in the linear integer programming approach). The exact time of leaving block sections is of secondary importance. Therefore, we consider those excited states that describe the same order of trains as the ground state to be equivalent to the actual ground state encoding the global optimum. As discussed in Section 3.3, our QUBO formulation problem ensures that those equivalent solutions are present in the low-energy spectrum.

#### 4.3.1. Exact Calculation of the Low-Energy Spectrum

To demonstrate the aforementioned idea, we first present the results of the brute-force numerical calculations performed on a GPU architecture [80]. With this approach, the spectra of the smallest instances have been calculated exactly, providing some guidance for the understanding of the model behavior and parameter dependence, especially with respect to the low-energy part of the spectrum. Note that the brute-force search is actually an enumeration of all the possible configurations computed using a GPU. The method is suitable for small (i.e., up to 50 quantum bits) but otherwise arbitrary systems. No embedding is needed. To study (hard) penalties resulting from non-feasible solutions, apart from $p_{\mathrm{pair}}=p_{\mathrm{sum}}=1.75$ in Equation (Equation 26), we use other higher penalties that are not equal to each other, $p_{\mathrm{pair}}=2.7$ and $p_{\mathrm{sum}}=2.2$.

Let us assume that the solution in Figure 3b is the optimal one. Here, the train IC3521 (w=1.5) waits 3 min at block 3, while regional train R90602 (w=1.0) waits 4 min at block 5, causing 4 min of secondary delay upon leaving block 3. This adds 1.214 to the objective. Concerning the feasibility terms in Equation (Equation 26), for a feasible solution $P_{\mathrm{sum}}^{\prime }=0$, while the linear constraint gives the negative offset to the energy. According to Equation (Equation 25), as we have three trains for which we analyze two stations, this negative offset is $P_{\mathrm{sum}}=-3\cdot 2\cdot p_{\mathrm{pair}}$. Based on the feasibility terms set out Equation (Equation 26), this yields −10.5 for $p_{\mathrm{sum}}=1.75$ and −13.2 for $p_{\mathrm{sum}}=2.2$. This results in a ground-state energy of $f^{\prime }\left(x\right)=-9.286$ and $f^{\prime }\left(x\right)=-11.986$, respectively. Finally, in the ground-state solution shown in Figure 3b, the IC3521 train can leave the station block 1 with a secondary delay of 0, 1, 2, or 3, not affecting any delays of the trains leaving block 3. All these situations correspond to the ground state energy. Hence, our approach produces a 4-fold degeneracy of the ground state.

Low-energy spectra of the solutions and their degeneracy are presented in Figure 4a,b. All the solutions that are equivalent to the ground state from the dispatching point of view are marked in green. Infeasible excited state solutions (in which some of the feasibility conditions set out in Equation (Equation 26) are violated) are marked in red. In this example, we do not have feasible solutions that are not optimal, i.e., in which the order of trains at a station is different from the one in the ground-state solution.

In the case of line No. 191, a more detailed analysis of the low-energy spectra of the solutions was possible due to the generality of the brute-force simulation. The results are presented in Figure 4. We shall find later on that the D-Wave solutions were in the “green” tail of feasible solutions, but the high degeneracy of higher-energy states may impose some risk of the quantum annealing ending up in the more frequently appearing excited states (see Figure 5).

#### 4.3.2. Classical Algorithms for the Linear (Integer Programming) IP Model and QUBO

We expect classical algorithms for QUBOs to achieve the ground state of Equation (Equation 26) or at least low excited states equivalent to the ground state with respect to the dispatching problem. It is important to mention that hereafter, we embed the original QUBO into the Chimera graph (see Section 2.3.1). This makes the algorithm ready for processing on a real quantum annealer.

As to a simple example of the embedding, we refer to the problem with four quantum bits that has been discussed in Appendix A. In that case, the mapping was trivial. In a case of six quantum bits, for instance (by setting $d_{\max}=2$), we will have additional terms in Appendix A. Hence, the larger problems cannot be directly mapped onto the Chimera graph, so the embedding procedure is required, as illustrated in Figure 6. This illustrates the basic idea of how the embedding is performed in even larger models.

As to the model parameters, recall that for the particular QUBO, we have opted for $p_{\mathrm{pair}}=p_{\mathrm{sum}}=1.75$ or $p_{\mathrm{pair}}=2.2\;p_{\mathrm{sum}}=2.7$ for line No. 216 and $p_{\mathrm{pair}}=p_{\mathrm{sum}}=1.75$ for line No. 191. Let us present the solutions of the two state-of-the-art numerical methods, which we shall later compare with the experimental results obtained by running the D-Wave 2000Q quantum annealers.

The first solver is developed ‘in-house’ and is based on tensor network techniques [76]. The solver is designed to efficiently sample high-quality solutions of certain spin-glass systems with the aim of solving hard optimization problems, and it has proven to be applicable in our case. The idea behind this solver is to represent the probability of finding a given configuration by a quantum annealing processor as a PEPS tensor network. This allows an efficient bound-and-branch strategy to be applied in order to find $M\ll 2^{N}$ candidates for the low-energy states, where *N* is the number of physical quantum bits on the Chimera graph. In principle, such a heuristic method should work well for rather simple QUBO problems, i.e., those in which the *Q* matrix in Equation (Equation 4) has some identical or zero terms; this corresponds to the so-called weak entanglement regime. It can be shown that this is the case in our problem (see also the simple example of the *Q* matrix in Appendix A), which makes the algorithm applicable in the present context. Furthermore, heuristic parameters such as the Boltzmann temperature (β) can be provided, allowing one to zoom in on the low-energy spectrum depending on the problem in question. We set β=4, which is quite a typical setting, as discussed in [76]. Although even better solutions may potentially be achieved by further tuning this parameter, we demonstrated that this default setting is satisfactory from the dispatching point of view. The second classical solver is CPLEX [86] (version 12.9.0.0). In our work, we have used the DOcplex Mathematical Programming package (DOcplex.MP) for Python. In what follows, “CPLEX” refers to the QUBO solver of this package.

In order to have a fair comparison with a traditional approach, we have also formulated our model as a linear integer program; this is described in Appendix A in detail. We have implemented the linear integer model with the PuLP package [87] and solved with its default solver (CBC MILP Solver Version: 2.9.0). All instances were solved to the optimal solution in 0.03 s on an average computer. This was in line with our expectations, as our problems are small. Our goal is, however, not to outperform either CPLEX or the standard linear solver but to demonstrate the applicability of quantum hardware; at the present state of the art, we need the well-established solvers to produce results for comparisons and reference.

Concerning the results of the other railway line (No. 191), the values of the objective function in Equation (Equation 27) are given in Table 2. We also include the values of our objective function for the FLFS, FCFS, and AMCC optimal solutions.

The agreement with the linear integer programming approach provides the argument that the CPLEX results correspond to the ground state of the QUBO. We are interested in the results being equivalent to those of CPLEX and the linear solver from the dispatching point of view. These results are marked in blue in Table 2. The tensor network approach yields equivalent solutions to those of CPLEX. However, the tensor network sometimes returns excited states of the QUBO, as can be observed in case 3. The reason for this is that the tensor network method is based on approximations. This demonstrates that even some low-energy excited states encode a satisfactory solution. Interestingly, the results of the AMCC are also equivalent to those CPLEX in cases 2, 3, and 4 but different from those in case 1. The reason is that the AMCC needs to have a specific objective function, whereas in our approach, we can choose this function more flexibly. Specifically, in case 1, the meet-and-pass situation of trains IC1 and Ks2 at station 10 yields the lowest maximum secondary delay, so it is optimal from the AMCC point of view. (Note that two trains have secondary delays: Ks2 and Ks3 in this case.) As discussed earlier, in this approach, Ks2 is prioritized, as it is the train leaving the modeled *network* segment, and one of the goals is to limit delays propagating further from this segment. The train diagrams based on the CPLEX solutions are depicted in Appendix A.

In case 3, observe that the objective function in Table 2 from the tensor network solution differs from the minimum (yet the solution is still equivalent to the optimal one). To explain this, observe that there are numerous possibilities of additional train delays that do not affect the dispatching situation. An example of such a situation is a train having its stopover extended at the station with no meet and pass or meet and overtake. Such a situation increases the value of the objective but does not affect the optimal dispatching solution. The number of combinations here is high, and this is why such extended stopovers may be returned by the approximate algorithm. This is in contrast to the exact FCFS, FLFS, and AMCC heuristics, which do not allow for such unnecessary delays; the exact heuristics always return f(x) that is the minimum for the particular dispatching solution. In case 3, the FCFS with f(x)=0.95 does not give the optimal solution from the dispatching point of view, as opposed to the tensor network with f(x)=1.65.

#### 4.3.3. Quantum Annealing on the D-Wave Machine

As described in Section 2.3.2, physical quantum annealers are probabilistic. In particular, as the required time to drive the system into its ground state is unknown, the output is a sample of the low-energy spectrum from repeated annealing processes, hence it can be regarded as a heuristic. The solution is thus assumed to be the element of this sample with the lowest energy (in practice, these elements are not from the ground states but from low excited states). The likelihood of obtaining solutions with a lower energy (or the actual ground state) increases with the number of repetitions.

As already mentioned, qubits on the D-Wave’s chip are arranged into a Chimera graph topology. Furthermore, some nodes and edges may be missing on the physical device, making the topology different even from an ideal Chimera graph. This requires *minor embeddeding* of the problem, mapping logical qubits onto physical ones. To this end, multiple physical qubits are chained together to represent a single logical variable, which increases their connectivity at the cost of the number of available qubits. Such embedding is performed by introducing an additional *penalty term* that favors states in which the quantum bits in each chain are aligned in the same direction. The multiplicative factor governing this process is called the chain strength, and it should dominate all the coefficients present in the original problem. (Note that we encounter yet another penalty term at this point.) In this work, we set this factor to the maximum absolute value of the coefficients of the original problem multipled by a parameter that we call the *chain strength scale* (css). In our experiment, *css* ranged from 2.0 to 9.0. Another parameter is the annealing time (ranging from 5 to 2000 μs). This is the actual duration of the physical annealing process.

The data flow after performing the calculation on the quantum hardware is the following. The raw solution returned by the hardware consists of the configurations of the physical spins or QUBO variables depending on the format of the submitted problem, the corresponding energies, multiplicities, and other technical parameters. Therefore, the solution has to be transformed to logical variables by reverting the embedding. The physical variables representing the same logical one have equal values ideally; however, in reality, a *chain break* can occur: some of them may have different values. These are resolved by “majority voting”. The transformation between Ising and QUBO variables and the conversion of the logical variables are all implemented in the software package supplied by D-Wave (even though the raw data are also available), so having submitted the QUBO, one obtains 0–1 solution vectors along with energies and multiplicities. From these, the *conflict free* timetable can be decoded, as already shown in Section 3.

In Figure 7c,d, we present the energies of the best outcomes of the D-Wave machine for line No. 216 and various annealing times. The green dots denote the feasible solutions (and equivalent to the optimal solution), while the red dots denote solutions that are not feasible. In general, the quality of a solution slightly improves with the annealing time; however, in large examples, the best results are achieved for a time between 1000 and 2000 μs. This coincides with the observation in [58], in which quantum annealing on the D-Wave machine was performed on various problems too, and it was demonstrated that for a moderate problem size, the performance (in terms of the probability of success) improves with an annealing time of up to 1000 μs. Hence, we have limited ourselves to the annealing times of the order of magnitude of 1000 μs in analyzing larger examples.

Rather counterintuitively, setting lower penalty coefficients of $p_{\mathrm{sum}}=p_{\mathrm{pair}}=1.75$ for the hard constraints resulted in samples containing more feasible solutions. For this reason, we had kept this penalty setting for the analysis of the larger case. The embedding strength was set to css=2 in this case, i.e., the lowest possible value. This has proven to be a good choice, as demonstrated in Figure 8. The best D-Wave solutions are presented in the form of train diagrams in Figure 7a,b.

The quality of the solutions in relation to the css strength in the various parameter settings is presented in Figure 8. We observed that in our cases, the quality of the solution degraded with an increase in css. This is unusual, as increasing the css strength typically yields more solutions without broken chains that do not need to be post-processed to obtain a feasible solution of the original problem. This may be caused by the fact that the large coupling of the embedding may cause the constraints to appear as a small perturbation in the physical QUBO. These perturbations, as discussed earlier, may be hidden in the noise of the D-Wave 2000Q annealer.

Hence, we set css=2.0 (the lowest possible value) for the further investigations. Some examples of the penalty and objective function values are presented in Table 3. Again, it appears that the higher the values of $p_{\mathrm{sum}}$ and $p_{\mathrm{pair}}$, the higher the values of f(x). This may be caused by the objective function being lost in the noise of the D-Wave 2000Q annealer.

For railway line No. 191, finding a feasible solution is more difficult. Hence, we increased the number of samples to 250,000. The results of the lowest energies and penalties are presented in Figure 9. We had to skip case 3 because the higher number of feasibility constraints prevented finding any embedding on a real Chimera. Interestingly, recall that we found the embedding for the ideal Chimera while simulating the solution (see Section 4.3.2). Hence, the failure in the case of the real graph is possibly due to the lack of certain required connections or nodes from the real Chimera. Finding the feasible solution in such a case (while having non-zero hard constraints penalties) is a problem for further research. One would expect that increasing the $p_{\mathrm{pair}}$ and $p_{\mathrm{sum}}$ parameters could be helpful. However, it may aggravate the objective function to an ever greater extent. In Figure 9b, the values of the objective function f(x) are much higher than the optimal ones presented in Table 2.

Although the solutions are not feasible, we can still select the two in which only one hard constraint is violated ($f^{\prime }\left(x\right)=1.75$); these are case 1 with an annealing time of 1400 μs and case 2 with an annealing time of 1200 μs. The train diagrams of these solutions are presented in Figure 10. Note that both these diagrams can easily be modified by the dispatcher to obtain a feasible solution. The case in Figure 10a can be amended by adding the lacking 1-min stay of Ks3 in station 7. This solution would not be optimal, and thus, it would be different from the optimal one obtainable with CPLEX, the tensor network solver, or FLFS. It would also differ from the non-optimal yet feasible ones returned by FCFS and AMCC. Similarly, the case in Figure 10b can be upgraded by shortening the stays of Ks3 and IC2 and letting them meet and pass at station 10. The so-obtained solution would be optimal.

At this point, a comment on the degeneracy of the ground state is in order. Clearly, in the instances related to the railway line 191, we are facing degenerate ground states. The reason for this is that in our model in Equation (Equation 27), the delay is penalized in the objective at the end of the trains’ routes. Hence, there are many possibilities for trains to wait on various stations to meet the dispatching conditions. These choices lead to the same ground-state energy. What quantum annealing provides is a sample of the low-energy spectrum, possibly involving some of these ground states. In our case, however, the just mentioned intuitive interpretation of the solution enables us to find a practically useful and close-to-optimal solution.

The real D-Wave quantum annealing is tied to some parameters of both the particular QUBO and the machine itself. In our experiments, we varied only the annealing time, number of performed reads and the chain strength scale (css), leaving all other parameters at their default values. We achieved the best results for a coupling constant css = 2.0 for the small example in Figure 8a; the same observation was made for the large example. This was not expected, as the coupling between quantum bits representing a single classical bit was rather weak. Here, we probably took advantage of the possible variations within the realization of a logical bit. This observation demonstrates that the embedding selection may be meaningful in searching for the convergence toward proper solutions lying in the low-energy part of the spectrum. For the small cases, we observed a feasible solution for a relatively small number of samples (equal to 1 k). For the larger case, we increased the number of samples to 250 k and still we did not reach any feasible solution. The conclusion is that the impact of the noise amplifies strongly with the size of the problem. The convergence of the best obtained solution toward the optimal one with the given sample size is complex, and an in-depth statistical analysis that samples the annealer’s real distribution is required.

As demonstrated in Figure 9b, in some cases, only a single hard constraint was violated. This may suggest that we are near the region of feasible solutions. However, the objective function values are still far from the optimal ones achieved by means of simulations (see Table 2). To elucidate the interplay between penalties, we refer to Figure 10, in which the solutions are not feasible but can be easily corrected by the dispatcher to obtain feasible ones. In Figure 10b the corrected solution would be optimal, while in Figure 10a, it would not be different from all the other achieved solutions. Hence, the current quantum annealer would rather sample the excited part of the QUBO spectrum, which can lead to unusual solutions. Such solutions, however, can be still be used by the dispatcher for some particular reason not encoded directly in the model. Such reasons include unexpected dispatching problems, rolling stock emergency, and non-standard requirements.

Let us also mention the characteristics of our QUBO problems as they are important features from the point of view of quantum methods. Table 4 summarizes the problem sizes and the densities of edges in the case of each problem instance.

### 4.4. Initial Studies on the D-Wave Advantage Machine

During the preparation of the present paper, a new quantum device, the D-Wave’s Advantage_system1.1 system (with an underlying topology code-named Pegasus [11]), became commercially available. Hence, we performed preliminary experiments with this new architecture to address a slightly larger example. To that end, we expanded our initial Goleszów–Wisła Uzdrowisko (line No. 191) problem instance to be 3 times bigger in size. Furthermore, we investigated nine trains in each direction.

The conflicts were introduced by assuming delays of 20, 25, or 30 min of certain trains entering block Section 1. The control parameters’ values $p_{\mathrm{sum}}=p_{\mathrm{pair}}=1.75$ and css=2 were not changed. As a result, the problem was mapped onto a QUBO with 594 variables and 5552 connections. The physical topology of the new system is different from that of its predecessor, so a different embedding was needed. This did not have any fundamental implications in our case; hence, we do not discuss its details here. Employing a strategy similar to the one used for our other calculations, we used the solution found by CPLEX as a reference for comparisons.

After performing 25 k runs, we reached a minimal energy of +75.28 with an annealing time of 1400 μs (the raw computational time on the D-Wave machine was 35 s). Unfortunately, this is not a feasible solution. The CPLEX calculations, on the other hand, resulted in an energy of −92.43 with an objective function value f(x)=2.07 (see Figure 11). This is the same solution as the solution of the linear solver obtained using COIN-OR in 0.02 s. This solution is substantially better, and as it coincides with the linear solver’s output, it corresponds to the ground state. Our preliminary experiments indicate the need for a more detailed investigation of the new device’s behavior (and that of the current model) to determine whether obtaining solutions with the desired (better) quality is possible. A part of this problem will likely be eliminated simply by the technological development of the new annealer. For an intuitive justification, we refer to [78] and Figure 1 therein, in which an improvement in the performance between subsequent iterations within one generation of Chimera-based quantum annealers was observed. As discussed in Section 3.2, the number of logical bits (variables) and number of edges (quadratic terms) scales roughly as numberoftrains · (numberofstations − 1) for constant $d_{\max}$. Using this approximation, and referring to ref Table 4, for twice as many trains as in Figure 11 (roughly whole day of operation), we would have approximately 1200 logical bits and 11,000 edges. If we further enlarge the problem to the whole branch line (n.o. 157, 190, and 191) with eight stations, there would be approximately 2800 logical bits and 26000 edges.

## 5. Discussion and Conclusions

The NISQ era [10] is here. Early generations of quantum computing hardware have become available that may serve as a stepping stone for the development of practically useful technologies that exhibit genuine quantum advantage. However, until such “real” quantum computers are available, it is imperative to demonstrate how and which real-world applications are amendable to be solved on quantum computing hardware. To this end, we have introduced a new approach to the single-track line dispatching problem that can be implemented on a real quantum annealing device (D-Wave 2000Q). Namely, we have addressed two particular real-life railway dispatching problems in Poland; many similar examples exist in other networks, too. Specifically, we have introduced a QUBO model of the problem that can be solved with quantum annealing, and we have benchmarked it against classical algorithms.

The first dispatching problem we considered (the Nidzica–Olsztynek section of line No. 216) was particularly small, and it was defined using only 48 logical quantum bits (which we were able to embed into 373 physical quantum bits of a real quantum processor). The final state reached by the quantum annealer for this problem was optimal for many parameter settings. This highlights that small-sized dispatching problems are already within reach of near-term quantum annealers. In addition, the limited size of the problem made it possible to analyze the QUBO with a greedy brute-force search algorithm, which revealed details of the behavior of the spectrum that cannot be exactly calculated for bigger instances.

Our second set of dispatching problems (the Goleszów–Wisła Uzdrowisko section of line No. 191) was larger and needed 198 logical quantum bits. Here, the number of physical quantum bits depends on the number of constraints in each of the analyzed cases. We were able to embed all four dispatching cases of the No. 191 railway line into an ideal Chimera graph (2048 physical quantum bits) using a state-of-the-art embedding algorithm. We succeeded in solving these instances with classical solvers for QUBOs. Meanwhile, on the physical device (whose graph is not perfect and lacks several quantum bits and couplings), we were able to embed only three out of the four cases (case 3, with the highest number of conflicts, could not be embedded). We expect that such obstacles will become less restrictive as new embedding algorithms are being developed for both the current Chimera topology and the newest D-Wave Pegasus; see [88,89]. Therefore, it is not unreasonable to expect that the range of problems that can be embedded so that they can be solved on physical hardware will substantially increase in the near future. Unfortunately, the D-Wave 2000Q solutions of our second problem appeared to be far from optimal. This is attributable to the noise that is still present in the current quantum machine.

We have successfully solved our model using several algorithms for QUBOs running on classical computers, notably the novel tensor network method. This introduces additional possibilities, namely, that of QUBO modeling and the use of quantum-motivated classical algorithms. Although these possibilities obviously do not promise a breakthrough in scalability, they are essential for the validation and assessment of the results of real quantum annealing. In addition, they can yield practically useful results.In fact, hybrid quantum-classical computing is a promising avenue of research, which has recently seen significant development [90,91].

We are aware that the examples of the single-track railway dispatching problem discussed in the paper can be regarded as trivial from the point of view of professional dispatchers. This is also reflected by the efficiency of the conventional linear solver they may use. Our intention, however, was to provide a proof-of-concept demonstration of the applicability of quantum annealing in this field. This goal has been achieved: we have described a suitable model and succeeded in solving certain instances.

Due to the small size of the current quantum annealing processors, our implementation is limited: quantum annealing is an emerging technology. Owing to the significant efforts put into the development of quantum annealers, the addressable problem sizes are about to increase, and the quality of the samples will also improve. With the development of the technology, it is not far-fetched to realize that at some point, soon quantum annealers will be able to compete with or even outperform classical solvers. In particular, hybrid quantum-classical algorithms applied to the here presented type of model may even reach the size and complexity of the limitations of the state-of-the-art classical algorithms.

## Figures and Tables

**Figure 1 entropy-25-00191-f001:**
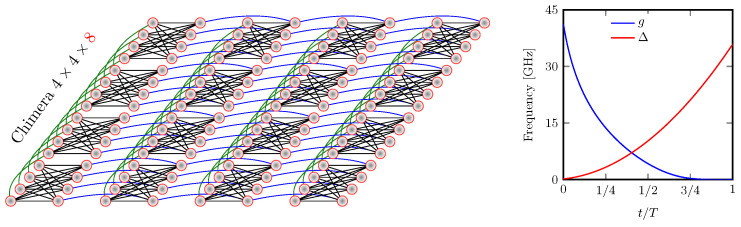
**D-Wave processor specification**. Left: An example of the Chimera topology, here composed of 4×4 ($C_{4}$) grid consisting of clusters (units cells) of 8 qubits each. The total number of variables (vertices) for this graph is N=4·4·8=128. A graph’s edges indicate possible interactions between qubits. The maximum number of qubits is $N_{\max}=2048$ for the Chimera $C_{16}$ topology, whereas the total number of connections between them is limited to $6000\ll N_{\max}^{2}$. Right: A typical annealing schedule controlling the evolution of a quantum processor, where *T* denotes the time to complete one annealing cycle (the annealing time). It ranges from microseconds (∼2 μs) to milliseconds (∼2000 μs). The parameters *g* and Δ are used in Equation (Equation 6).

**Figure 2 entropy-25-00191-f002:**
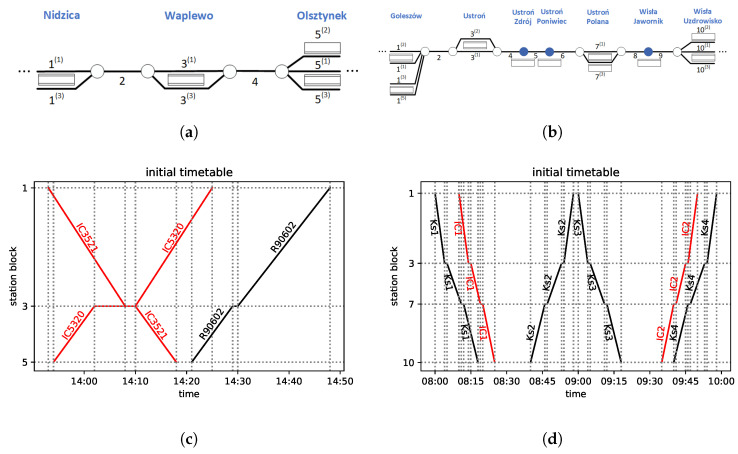
The railway line segments and their initial (undisturbed) timetables addressed in our calculations. The train diagrams in subfigures (**c**,**d**) represent train paths by connecting characteristic points of the location of trains at certain times by straight lines. Subfigures (**a**,**b**) represent the *network* topologies. The lines are the railway tracks. Their numbers represent blocks (as used, e.g., in the vertical axes of the train diagrams) and their upper indices in parentheses refer to the sidings (i.e., parallel tracks of stations). The rectangles represent the passenger platforms, circles represent the block boundaries (white: between a station and a line block, blue filled: between two line blocks). (**a**) Nidzica–Olsztynek section of railway line No. 216. (**b**) Goleszów–Wisła Uzdrowisko section of railway line No. 191. (**c**) Train diagram for the timetable of the line in (**a**). (**d**) Train diagram for the timetable of the line in (**b**).

**Figure 3 entropy-25-00191-f003:**
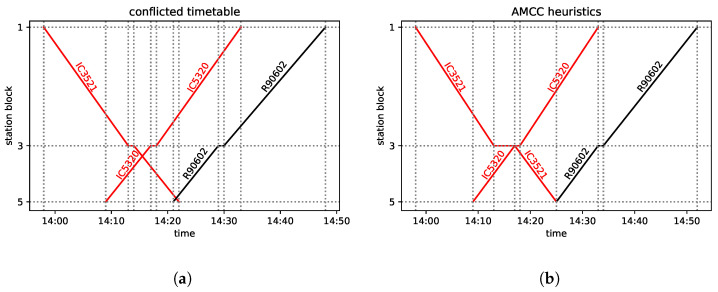
A possible solution of the conflict on line No. 216. (**a**) The conflicted diagram. All the three trains would meet in block 4 as it can be seen from the intersecting train paths. (**b**) The solution; FCFS, FLFS, and AMCC give the same outcome with a maximum seconday delay of 4 min.

**Figure 4 entropy-25-00191-f004:**
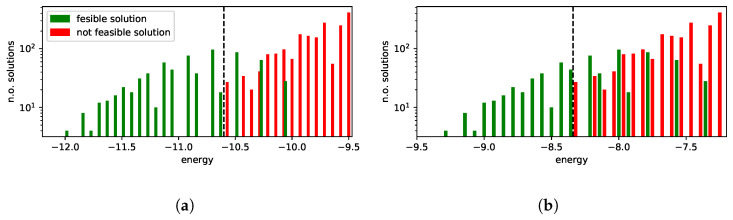
Spectra of the low-energy solutions for two penalty strategies of the brute-force (exact) solution. The black line separates the phase in which only feasible solutions appear. Observe the mixing phase, in which both feasible and unfeasible solutions occur. Here, $p_{\mathrm{pair}}$ and $p_{\mathrm{sum}}$ are penalties of the unconstrained problem expressed in the “logical” variables. The term $p_{\mathrm{sum}}={\left(\sum_{i\in V_{s}}x_{i}-1\right)}^{2}$, cf. Equation (Equation 24), ensures that each train leaves a station only once, whereas $p_{\mathrm{pair}}=\sum_{\left(i,j\right)\in V_{p}}\left(x_{i}x_{j}+x_{j}x_{i}\right)$, cf. Equation (Equation 25), imposes the following: minimal passing time constrain, single block occupation constrain, deadlock constrain, and rolling stock circulation constrain. (**a**) $p_{\mathrm{pair}}=2.7,p_{\mathrm{sum}}=2.2$. (**b**) $p_{\mathrm{pair}}=p_{\mathrm{sum}}=1.75$.

**Figure 5 entropy-25-00191-f005:**
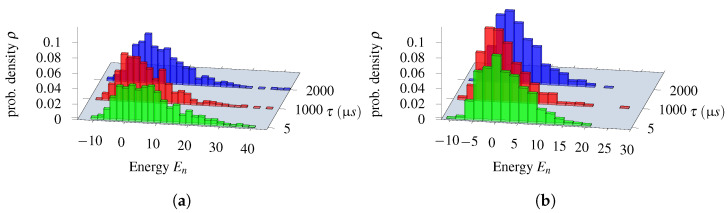
Distribution of the energies corresponding to the states (solutions), which are sampled by the D-Wave 2000Q quantum annealer of 48 logical quantum bits instance of line No. 216. In particular, 1000 samples were taken for each annealing time, and the strength of embedding was set to css=2.0. This device is still very noisy and prone to errors, so the sample contains excited states. (**a**) QUBO param.: $p_{\mathrm{pair}}=2.7,p_{\mathrm{sum}}=2.2$. (**b**) $p_{\mathrm{pair}}=p_{\mathrm{sum}}=1.75$.

**Figure 6 entropy-25-00191-f006:**
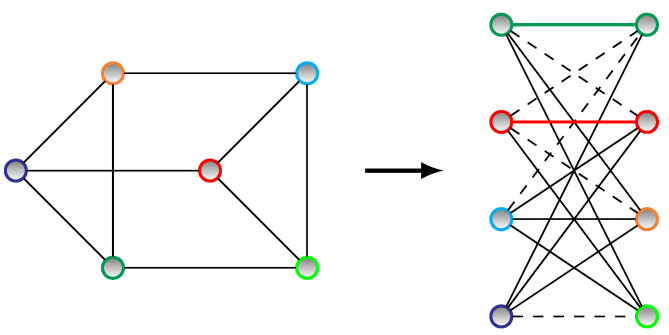
Embedding of a simple, six-qubit problem. (**Left**) graph of the original problem. (**Right**) problem embedded into a unit cell of Chimera. Here, different colors correspond to different logical variables. Apparently, the original problem does not map directly onto Chimera as it contains cycles of length 3. Therefore, two chains have to be introduced. Couplings corresponding to inner-chain penalties are marked with the same color as the variable to which they correspond.

**Figure 7 entropy-25-00191-f007:**
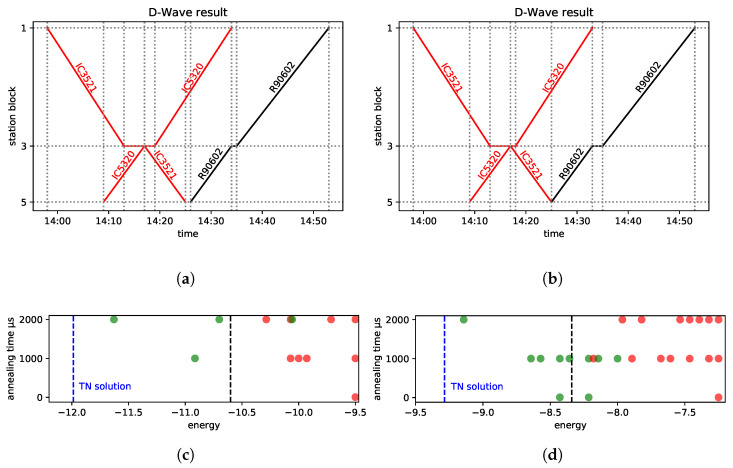
Train diagrams of the best D-Wave solutions, the lowest energies of the quantum annealing on the D-Wave machine (green: feasible, red: not feasible), and the optimal tensor network solution. The raw computational time on the D-Wave (n.o. runs × annealing time) was in the range $5\times 10^{-3}$–2 s. (**a**) The optimal solution from (**c**). (**b**) The optimal solution from (**d**). (**c**) $p_{\mathrm{pair}}=2.7,p_{\mathrm{sum}}=2.2$. (**d**) $p_{\mathrm{pair}}=p_{\mathrm{sum}}=1.75$.

**Figure 8 entropy-25-00191-f008:**
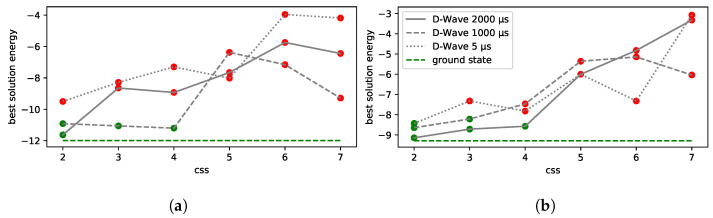
Line No. 216, with the minimal energy from the D-Wave quantum annealer, using 1000 runs. Green dots indicates the feasible solutions, while the red dots denote the unfeasible ones. In general, the energy rises as the css strength rises. We do not observe that the different settings of $p_{\mathrm{pair}}$ and $p_{\mathrm{sum}}$ improve the feasibility; see (**a**) The minimal energies vs. css for $p_{\mathrm{pair}}=2.2,p_{\mathrm{sum}}=2.7$. (**b**) The minimal energies vs. css for $p_{\mathrm{pair}}=1.75,p_{\mathrm{sum}}=1.75$.

**Figure 9 entropy-25-00191-f009:**
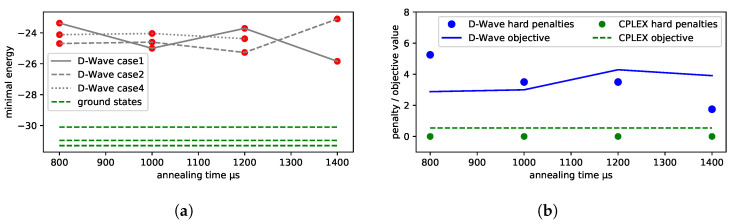
Line No. 191, with the minimal energy from the D-Wave annealer at 250 k runs, css = 2.0, and $p_{\mathrm{pair}}=1.75,p_{\mathrm{sum}}=1.75$. The output does not dependent on the annealing time (in the investigated range) and is still far from the ground state. The raw computational time on the D-Wave (n.o. runs × annealing time) was in the range 200–350 s. (**a**) Best D-Wave solutions (these are the lowest excited states we have recorded). Red dots indicate that the solutions are not feasible. (**b**) Comparison of the objective and hard penalty for the D-Wave outcome and the optimal solution calculated with CPLEX.

**Figure 10 entropy-25-00191-f010:**
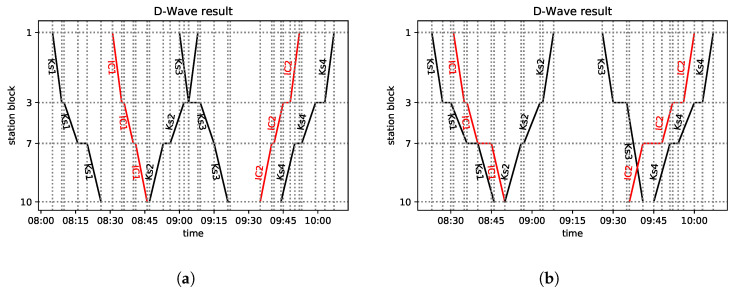
The best solutions obtained from the D-Wave quantum annealer for line No. 191. For case 1 (**a**), the annealing time is t=1400. The solution is unfeasible since the stay of Ks3 at station 7 is below 1 min. If the solution is corrected (i.e., the stay is introduced), it loses its optimailty and reflects a dispatching situation different from those obtained from FCFS, FLFS, AMCC, CPLEX, or the tensor network. For case 2 (**b**), t=1200 is used. The solution is unfeasible as Ks3 does not stop at station 7; hence, Ks3 and IC2 are supposed to meet and pass between stations 7 and 10. It can, however, be amended to an optimal solution: shortening the stay of Ks3 at station 3 and shortening the stay of IC2 at station 7 (and 3 if necessary) result in a meet-and-pass situation at station 10, and this is optimal. (**a**) Case 1. (**b**) Case 2.

**Figure 11 entropy-25-00191-f011:**
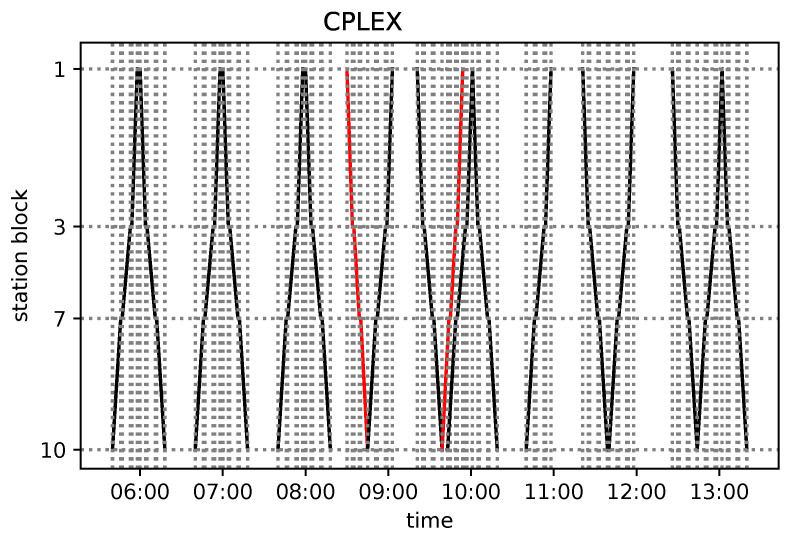
The CPLEX QUBO solution, coinciding with the linear model’s solution of the 18-train problem.

**Table 1 entropy-25-00191-t001:** The maximum secondary delays, in minutes, resulting from simple heuristics. Observe that for each case, there are solutions far below $d_{\max}=10$.

Heuristics	Case 1	Case 2	Case 3	Case 4
FLFS	6	13	4	2
FCFS	5	5	5	2
AMCC	5	5	4	2

**Table 2 entropy-25-00191-t002:** The values of the objective function f(x) for the solutions obtained by the classical calculation of the QUBO, linear integer programming approach, and all the heuristics. The blue color denotes equivalence from the dispatching point of view with the ground state of the QUBO or the output of the linear integer programming. The equivalence concerns the same order of trains at each station.

Method		Case 1	Case 2	Case 3	Case 4
QUBO approach	CPLEX	0.54	1.40	0.73	0.20
	tensor network	0.54	1.40	1.65	0.29
linear integer programming		0.54	1.40	0.73	0.20
Simple heuristics	AMCC	0.77	1.30	0.73	0.20
	FLFS	0.54	1.71	0.73	0.20
	FCFS	0.77	1.30	0.95	0.20

**Table 3 entropy-25-00191-t003:** Line No. 216, with the objective functions and penalties for violating the hard constraints: see Appendix A. Output from the D-Wave quantum annealer for the annealing time of 2000 μs. If $f^{\prime }\left(x\right)>0$, the solution is not feasible. The $p_{\mathrm{sum}}=p_{\mathrm{pair}}=1.75$ policy gives lower objectives.

css	$p_{\mathrm{sum}},p_{\mathrm{pair}}$	Hard Constraints’ Penalty $f^{\prime }\left(x\right)$	f(x)
2	1.75,1.75	0.0	1.36
2	2.2,2.7	0.0	1.57
4	1.75,1.75	0.0	1.93
4	2.2,2.7	2.2	2.07
6	1.75,1.75	5.25	0.43
6	2.2,2.7	6.6	0.86

**Table 4 entropy-25-00191-t004:** Graph densities for various problems. Since case 3 is supposed to be the most complicated one of cases 1–4, it has the largest graph density, #—number of.

Features	Line 216	Line 191				
		Case 1	Case 2	Case 3	Case 4	Enlarged
problem size (# logical bits)	48	198	198	198	198	594
# edges	395	1851	2038	2180	1831	5552
density (vs. full graph)	0.35	0.095	0.104	0.111	0.094	0.032
embedding into	Chimera	Chimera	Chimera	Ideal Chimera	Chimera	Pegasus
approximate # physical bits	373	<2048	<2048	≈ 2048	<2048	<5760

## Data Availability

The authors will supply data for any reasonable demand.

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
