# Peer review of "Quantum Annealing in the NISQ Era: Railway Conflict Management"

_entropy, 2023, doi:10.3390/e25020191_

Round 1

Reviewer 1 Report

The authors discuss the use of quantum annealing for solving a real life computational problem: “delay and conflict management on single-track railway lines“. The analysis is very detailed and convincing. The work emphasis is somehow pedagogical since the work aims to be read by different communities. After mapping the problem into a Ising-like form, as usual in optimisation problems, some instances are verified by using a D-wave computers. The results are interesting even if they are not able to prove a supremacy of the quantum annealing compared to classical algorithms. The work is interesting the analysis complete and, in my opinion, the paper deserves to be published. There is only one point the authors should clarify better. When they analyse how other algorithms work compared to the one proposed the cite "sophisticated algorithm based on tensor networks". It does not emerges clear how such a class of algorithms can be use in the present context and I am skeptical on this possibility. A discussion on this question would be welcomed.

Author Response

We are grateful, for this remark. We made the issue of "sophisticated algorithm based on tensor
networks" a bit unclear. What we had meant about it was PEPS tensor network algorithm in Ref. [76], developed by the part of the authors of the present work in order to efficiently sample the low-energy spectrum of certain spin-glass systems, including the Ising models arising in the present context. We have modified the sentence in the abstract to make it clear. Further we have revised 4.3.2. to further emphasize why the given algorithm is really applicable in the present context.
And indeed, as the results demonstrate, the algorithm did work; the results of tensor networks outputs are presented  in section 4.3.2 Tab 2. 
This approach to QUBO gave solutions that are equivalent (from the dispatching point of view) with the CPLEX approach to QUBO, and ILP approach. We hope that we have succeeded in eliminating the doubts of the referee.

Reviewer 2 Report

The authors have presented a detailed study on railway conflict management problems with quantum annealing, exemplifying their protocol through real-life cases in the Polish railway network. To be more specific, they reformulate the problem into solving the ground state of an Ising Hamiltonian, which encodes the solution in spin configurations. They have also tested their protocols in the D-Wave quantum annealer and benchmarked the results with classical algorithms. Overall, the manuscript is scientifically sound and clearly written. I will recommend this interdisciplinary work for publication if the authors address my questions as follows.

  1. As the authors have mentioned, the QUBO formulation requires introducing constraint conditions into the cost function with Lagrangian multipliers, for which the penalty setting is tricky. I am wondering how they choose p_sum and p_pair instead of trying from tiny values, which gives rise to additional algorithm complexity. A description of their methodology will be welcomed.
  2. It would be great if the authors could briefly analyze the performance of the quantum annealing when the ground state degenerates, i.e., there are equivalent solutions with the same value of cost functions. It is quite common in problems for quantum annealing。

[Minor comment]

1. The order of figures might be altered based on their appearance, e.g., Fig. 2c and 2d be Fig. 1a and 1b.

2. There is a typo in footnote 2, which should be “Turing machine”.

Author Response

  1. Certainly it is possible to apply systematic approaches to determine the penalty parameters. (In case of linear constraints, for instance, it proven that it is always possible to find penalties that separate between feasible and infeasible configurations, as mentioned in the newly added Ref. [84] of the manuscript.) Still, determination of the parameters is a challenge. In our case, as we had numerical experience with the behavior of the objective as well as the penalties, 
    the actual values of parameters p_sum and p_pair were selected on the basis of this experience in an ad hoc manner.
    We had the intention to avoid the situation when the change in the objective due to the delay of a single train exceeds the constraint penalty. We have made comments on this in the manuscript, too.
  2. In the cases on railways line 191 we are facing degenerate ground states. The reason for this is that in our model in Eq. (27), the delay is penalized in the objective at the end of trains' routes. Hence, there are many possibilities for trains to wait on various stations to meet the dispatching conditions. These choices lead to the same ground state energy. What quantum annealing provides is a sample of the low energy spectrum, possibly involving some of these ground states. In our case, however, the just mentioned intuitive interpretation of the solution enables us to find a practically useful and close-to-optimal solution.  We have added this comment to the last part of Section 4.3.3 (page 24) where these examples are described. 
  3. The minor remarks have been corrected